# Human Cancers Derived from Either Genetic or Lifestyle Factors Are Initiated by Impaired Estrogen Signaling

**DOI:** 10.3390/cancers18010078

**Published:** 2025-12-26

**Authors:** Zsuzsanna Suba

**Affiliations:** Department of Molecular Pathology, National Institute of Oncology, Ráth György Str. 7-9, H-1122 Budapest, Hungary; subazdr@gmail.com; Tel.: +36-1-224-8600

**Keywords:** alcohol consumption, DNA damage, DNA repair, endocrine disruptor, estrogen, estrogen receptor, growth factor receptor, immune reaction, lifestyle factors, mutation, smoking

## Abstract

Impaired estrogen signaling caused by either endogenous or environmental factors alarms the hypothalamus, which then issues neural and hormonal commands throughout the whole body to restore genomic stability. In the first, compensated phase of defective estrogen signaling, patients appear healthy. Hyperinsulinemia is a subtle signal of the genome that indicates successful restoration of estrogen signaling at the expense of excessive insulin synthesis. Over time, the strengthening defect in estrogen signaling and impaired glucose uptake result in metabolic syndrome, with various symptoms and altered laboratory findings. The compensatory molecular changes are warnings from an endangered genome struggling to restore estrogen signaling and DNA stability. The worsening defect in estrogen signaling leads to type-2 diabetes, reflected by increased serum glucose levels despite all compensatory actions, while organs suffer from the lack of glucose and excessive lipid deposition. The complete breakdown of estrogen signaling results in DNA damage and cancer initiation in the affected organ. Cancer development is the genome’s cry for help to combat against dysregulation.

## 1. Introduction

Cancer is a major global challenge, attributed to its rapidly increasing morbidity and mortality rates worldwide [1]. The exact mechanism of cancer initiation and progression remains unclear, and the results of extensive therapeutic efforts are fairly questionable.

According to the reigning opinion, cancer initiation derives from the incidence of oncogenic mutations that may either be inherited, increasing susceptibility, or acquired, arising under harmful environmental exposure [2]. An accumulation of mutations in crucial genes presumably disrupt normal genomic regulation, leading to unrestrained cell proliferation. The majority of cancers may develop through unfortunate interactions between genetic predispositions and environmental influences [3]. Since only a small percentage of cancers exhibit solely inherited origin, small genetic differences among individuals may influence their response to environmental exposures, leading to increased or decreased risk for cancer.

Analysis of a large number of cancer genomes resulted in around 300 genes that were mutated in at least one type of cancer [4]. The exact roles of presumably oncogenic mutations in cancer development have not been clarified. Increased prevalence of certain somatic mutations in the genes of various cancers does not necessarily confirm their oncogenic impact; instead, it may reflect a genome’s effort to repair [5].

Historically, the first causal factor for cancer specifically affecting the female breast was introduced at the end of the 19th century. The clinically observed pulsation of tumors in parallel with the menstrual cycle mistakenly suggested that breast cancer growth may be associated with the ovulatory peak of serum estrogen levels. High concentrations of estrogen have emerged as a unique causal factor for premenopausal breast cancer. Surgical oophorectomy was applied as a causal therapy for breast cancer, targeting ovarian estrogen synthesis [6]. However, the regression of breast tumors via surgical estrogen withdrawal was modest and transient [7].

The therapeutic blockade of estrogen signaling has remained a gold standard for breast cancer care for centuries and continues to this day. However, the concept of estrogen-induced breast cancer led to further problems in cancer research and in the development of cancer therapy [8].

In 1988, a causal factor for human diseases (and later for cancer) was introduced into the scientific world. Gerald Reaven established and popularized his concept in his Banting Lecture that a defect of insulin-assisted glucose uptake, known as insulin resistance, is the source of many human diseases [9]. Insulin resistance was linked with a cluster of metabolic disorders, summarized as metabolic syndrome or syndrome X. Metabolic syndrome is characterized by impaired glucose metabolism, high blood pressure, central obesity, and low HDL-cholesterol coupled with increased triglyceride levels [10].

In the early stage of insulin resistance, compensatory hyperinsulinemia maintains the blood glucose levels within the physiological range by stimulating increased insulin secretion from pancreatic β-cells [11]. Advanced insulin resistance leads to metabolic syndrome, obesity, and non-alcoholic fatty liver disease [12]. Type-2 diabetes develops when the compensatory insulin synthesis cannot keep up with the insulin demand of the insulin-resistant target cells, leading to hyperglycemia [13]. Thorough investigations have equivocally revealed that insulin resistance is the underlying disorder for developing cardiovascular diseases and cancers, which are the leading causes of human mortality [14,15]. Since insulin resistance leads to a significantly wide range of chronic human conditions, clarifying its origin appears to be crucial for identifying the principal regulator of the human genome.

Insulin-resistant status shows gender-related differences, as men exhibit a significantly higher prevalence of metabolic diseases compared to women. In addition, among women, premenopausal cases are protected from impaired glucose uptake via their active hormonal cycles, whilst postmenopausal cases exhibit an incidence rate of insulin resistance very similar to age-matched men [16]. These gender- and menopause-related differences in the prevalence of insulin-resistant cases suggest that estrogen plays a crucial role in glucose homeostasis and energy balance [17]. Moreover, estrogen deficiency or impaired estrogen receptor activation appeared to be a causal factor for insulin resistance and the development of metabolic diseases [18,19].

Gender-related differences are also evident in the prevalence of cardiovascular diseases [20]. Healthy premenopausal women are typically protected from cardiovascular diseases and hypertension compared to age-matched men, suggesting protective effects of cycling estrogen levels [21]. Estrogen plays a crucial regulatory role in the health of the cardiovascular system [22]. After menopause, decreasing estrogen levels and the associated insulin resistance are important causal factors in the pathogenesis of atherosclerosis, myocardial dysfunction, cardiac hypertrophy, heart failure, and myocardial ischemia.

Pan-cancer analyses support that men, in general, show higher cancer incidence and worse survival than women across most cancer types [23]. These sex-based differences in cancer incidence are likely due to a combination of biological, environmental, and behavioral factors; however, robust estrogen signaling in women seems to be protective against cancer. Oral cancer incidence shows obvious gender-related differences [24]. Premenopausal women are highly protected from this tumor, while after menopause, estrogen loss is associated with a steeply increasing prevalence of the disease, even in nonsmoker, non-drinker cases. Breast cancer incidence is significantly higher in postmenopausal women as compared with premenopausal cases, showing increasing risk with age after menopause [25]. In postmenopausal women, the loss of estrogen increases the risk for type-2 diabetes and obesity; these chronic conditions are strong risk factors for breast cancer.

The mechanism by which insulin resistance induces oncogenes remains highly debated. Investigators largely follow the traditional theory suggesting that increased concentrations of certain molecular players and somatic mutations of their genes are causal factors for cancer initiation and progression. Over the past decades, hyperinsulinemia [15], hyperglycemia [26], overexpression of pro-inflammatory cytokines [27], and increased growth factor signaling [28] have been proposed as cancer initiators in insulin-resistant cases. Although the presence of these alterations is characteristic of insulin resistance, they may be regarded as compensatory efforts for restoring genomic functions rather than facilitating cancer development [8].

This work introduces defective estrogen signaling as a causal factor in cancer initiation and progression, highlighting the noteworthy genome-repairing efforts of healthy cells, tumor cells, and the whole human body.

## 2. Theories Suggesting Various Initiators of Insulin Resistance

There has been continuous debate on the origin of insulin resistance and the cause–effect relationship between insulin resistance and hyperinsulinemia (Table 1). It has been equivocally established that their coexistence leads to serious chronic conditions and increasing mortality [29].

Until the beginning of the 21st century, insulin resistance itself was regarded as the origin of disorders leading to metabolic syndrome and type-2 diabetes, while the associated excessive insulin synthesis seemed to be a compensatory action helping cellular glucose uptake [30]. The origin of insulin resistance appears to be variable, and the relative importance of each presumed causal factor has been thoroughly investigated [31]. In searching for the causal factors of insulin resistance, authors have focused on the roles of various pathological findings that are associated with impaired glucose uptake.

Low-grade inflammation is regarded as a causal factor of insulin resistance. It can disrupt insulin signaling pathways, leading to a reduced ability of cells to respond to insulin and to take up glucose from the blood [32]. Alterations in lipid metabolism, such as increased free fatty acids and the accumulation of lipid intermediates, cause insulin resistance by interfering with insulin signaling pathways. This lipotoxicity can impair cellular glucose uptake and leads to lipid deposition in tissues such as muscle and liver [33]. The gut microbiome influences insulin sensitivity by producing metabolites that can affect insulin sensitivity, energy metabolism, and inflammation. Lower microbial diversity and the presence of certain bacteria-producing inflammatory molecules are linked to insulin resistance. Conversely, higher diversity and the presence of specific bacteria, such as *Coprococcus*, may improve insulin sensitivity [34].

In the development of insulin resistance, both mitochondrial oxidative stress and mitochondrial dysfunction have been implicated. Experimental studies on insulin-resistant 3T3-L1 adipocytes from mice provided evidence that mitochondrial oxidative stress is the origin of insulin resistance, while oxidative phosphorylation remained unaffected [35]. The insulin resistance of muscle and adipose tissue appeared to originate from impaired intracellular GLUT4 transport [36], which could contribute to the development of metabolic changes and insulin resistance syndrome.

Recent studies have established that hyperinsulinemia equivocally precedes insulin resistance, becoming a causal factor in defective glucose uptake, obesity, type-2 diabetes, cardiovascular diseases, and cancer [37,38,39]. In PCOS (polycystic ovary syndrome) cases, hyperinsulinemia appears to be the predecessor of insulin resistance, creating a self-generating cycle that worsens reproductive and metabolic defects [43]. However, the initiator of either insulin resistance or hyperinsulinemia remains unclear.

Recent evidence suggests that genetic background, together with over-nutrition, may induce hyperinsulinemia, which appears to precede insulin resistance. Hyperinsulinemia is regarded as a major driver of obesity, type-2 diabetes, and their associated chronic conditions [44,45]. Presumably, energy-rich food intake drives increased insulin synthesis by pancreatic β-cells. Over time, these β-cells become exhausted, and their insulin production cannot keep up with the insulin demand of resistant cells, resulting in elevated blood sugar levels and type-2 diabetes. It has also been proposed that hyperinsulinemia dysregulates the balance of the insulin-GH-IGF axis, promoting lipid deposition.

The development of type-2 diabetes via hyperinsulinemia is explained by a proposed interplay among insulin, GH, and IGF-I [40]. Type-2 diabetes develops as increased insulin levels in the portal vein increase the liver’s sensitivity to growth hormone (GH). The liver then increases insulin-like growth factor-1 (IGF-1) production, which leads to negative feedback on the hypothalamus, significantly decreasing GH levels. The resulting low GH/insulin ratio stimulates lipogenesis and increases body weight.

In clinical practice, diabetes treatment using insulin with high affinity for IGF-I receptors improves the lipid profile, increases insulin sensitivity, and enhances glucose metabolism [41]. IGF-I showed cardio-protective effects and improved insulin sensitivity in patients following a myocardial infarction [42]. These results suggest that insulin-stimulated IGF-I signaling improves endocrine regulation and metabolic balance instead of disrupting them.

## 3. The Origin of Insulin Resistance Is a Defect in Estrogen Signaling, While Hyperinsulinemia Is a Compensatory Effort to Improve Estrogen Regulation

Since glucose is essential for cellular health and fuels all cellular functions, glucose deprivation significantly affects the continuous operation of the main regulatory circuits of ER alpha. These circuits control genome stabilization, cell growth/proliferation, and glucose supply [46]. Insulin and IGF-I are important players in the regulatory circuits of ERs, controlling glucose supply and metabolic balance. Impaired ER expression/activation or low estrogen levels upregulate compensatory insulin secretion, leading to hyperinsulinemia as an attempt to restore estrogen signaling, even if this effort is not successful.

Impaired estrogen signaling creates an emergency situation for mammalian cells by disrupting glucose and lipid metabolism while simultaneously endangering genomic stability (Table 2). The hypothalamus–hypophysis unit then sends neuro-endocrine signals to adipose tissue, directing the upregulation of estrogen synthesis and ER activation. Restoration of estrogen signaling improves glucose uptake, metabolic homeostasis, and energy expenditure in insulin-resistant patients [47,48].

Insulin and estrogen (specifically 17β-estradiol) signaling work together to activate glucose transporter 4 (GLUT4) by promoting its translocation to the plasma membrane [49]. Incorporation of GLUT4 into the plasma membrane allows the entrance of glucose into the cell. In estrogen deficiency, increased insulin levels may exert a complementary effect on GLUT4 translocation, facilitating cellular glucose uptake.

Insulin-like growth factor-1 (IGF-1) has a direct role on glucose uptake, working in partnership with insulin, particularly in peripheral sites, such as adipose tissue and muscles [50]. IGF-I promotes glucose uptake by activating its own IGF-I receptors and hybrid receptors that react to both IGF-I and insulin receptors (IRs). Amplified insulin levels facilitate IGF-I signaling. Then, IGF-I activation increases the unliganded activation of membrane-associated ERs, closing the regulatory circuit [51].

Low-grade inflammation is a characteristic finding in insulin-resistant adipose tissue. Insulin and IGF-1 signaling have complex regulatory effects on the development of inflammation. In healthy adipose tissue, they act as anti-inflammatory agents. Conversely, in insulin resistance, including obesity and aging, increased insulin and IGF-1 signaling together promote inflammation [52]. In insulin resistance, macrophages and immune competent cells are recruited to increase pro-inflammatory cytokine expression, facilitating aromatase enzyme activity and estrogen synthesis [53]. When the estrogen concentration reaches an adequate level, estrogen signaling improves cellular glucose uptake. Restored glucose uptake then reduces inflammation and lowers the estrogen concentration [8]. In insulin-resistant adipose tissue, low-grade inflammation remains, as estrogen levels cannot reach a suitably high concentration to improve glucose uptake.

Estrogen and insulin exhibit close interplay in regulating the balance between lipolysis and lipogenesis. In insulin resistance, impaired estrogen signaling liberates free fatty acids (FFAs), while hyperinsulinemia cannot exert its powerful anti-lipolysis effect [54]. In obesity, abundant FFAs in the circulation are pathologically deposited in non-adipose organs and tissues, such as the liver, pancreas, skeletal muscles, and the heart [55].

In postmenopausal women, estrogen deficiency decreases the body’s ability to oxidize (burn) lipids, leading to insulin resistance and fat accumulation. In obesity, weak estrogen signaling in the background leads to dysregulation of fatty acid metabolism and visceral lipid deposition [56]. Phosphodiesterase 3B (PDE3B) is an enzymatic hydrolyser of cAMP and cGMP pathways in cooperation with estrogen. They co-regulate the key metabolic actions of estrogen: lipolysis, energy homeostasis, and insulin secretion. Pharmacological blockade of AKT phosphorylation and PDE3B expression promoted lipolysis even in the presence of insulin [59]. These findings support that, in insulin resistance, estrogen deficiency-induced damage to AKT and PDE3B pathways inhibits the anti-lipolysis action of insulin, despite its high concentrations.

Takeuchi et al. published in Nature that in humans, increased carbohydrate metabolism by the gut microbiota contributes to the development of insulin resistance [60]. In insulin-resistant patients, increased fecal carbohydrate content, associated with microbial metabolism, showed strong correlations with inflammatory cytokines. Recently, it was established that gut bacteria associated with insulin sensitivity and insulin resistance exhibited different patterns of carbohydrate metabolism [61]. Additionally, bacteria associated with insulin sensitivity improved insulin resistance in a mouse model.

Unique bacterial components of the gut microbiome, known as the estrobolome, play a crucial role in the reactivation of inactive, bound estrogens, supporting insulin synthesis in pancreatic β-cells [57]. Inactive, conjugated estrogens delivered from the liver via bile secretion may be reactivated in the gut by estrobolome bacteria showing high β-glucuronidase activity. In conclusion, the accumulation of gut bacteria with high β-glucuronidase activity improves insulin sensitivity by increasing the levels of bioactive estrogens. In postmenopausal women, a partnership was observed between estrogen deficiency and dysbiosis of gut bacteria, leading to the development of type-2 diabetes [62].

Recently, gut bacterial sequences were found in healthy brain samples, suggesting that there is a microbiome in the brain [63]. Considering the importance of high β-glucuronidase activity in the gut, it can be presumed that intestinal bacteria also reactivate conjugated estrogens in the brain, increasing the levels of bioactive estrogen.

Mitochondrial dysfunction is a key feature of insulin resistance. Dysfunction of mitochondria leads to decreased β-oxidation, low ATP creation, and increased ROS production [64]. Estrogen maintains mitochondrial function by regulating mitochondrial respiration, mitochondrial biogenesis, and mitochondrial quality control [65]. Insulin and estrogen signaling converge on Sirt1, mTOR, and PI3K signaling pathways to jointly regulate autophagy and mitochondrial metabolism [58]. In insulin resistance, hyperinsulinemia serves as an effort to compensate for the lack of estrogen regulation on mitochondria.

In conclusion, estrogen signaling as the chief regulator of metabolic processes shows strong interplay with insulin, IGF-I, and other molecular players participating in the control of glucose uptake. The defect of estrogen signaling impairs glucose uptake while recruiting insulin and IGF-1 to restore the regulation of glucose and lipid metabolism. In crisis situations, growth factor receptors, including the IGF-1 receptor, are capable of the unliganded activation of ERs.

## 4. Impaired Estrogen Signaling Is the Origin of Genomic Instability and Insulin Resistance in BRCA Gene Mutation Carriers

Identification of the *BRCA1* and *BRCA2* genes was an enormous step in cancer research [66,67]. The protein products of these genes, BRCA1 and BRCA2, proved to be safeguards of genomic integrity, controlling transcriptional processes, DNA replication and recombination, and the repair of DNA damage [68].

Inheritable mutations of *BRCA1* and *BRCA2* genes increase the risk of hereditary cancers by causing genomic instability, particularly in female breasts and ovaries [69,70]. Well-functioning BRCA proteins play a pivotal role in the genomic stability of all cell types in men and women. However, the specific risk for breast cancer development in *BRCA1* mutation carriers, mistakenly suggested an underlying excess of estrogen signaling.

The receptor landscape of *BRCA1* mutant breast cancers shows typical ER-alpha negativity. Histologically, they are poorly differentiated, while clinically, they exhibit rapid growth and poor prognosis [71]. The vast majority of *BRCA1* mutant cancers are ER-alpha negative and ER-PR-HER2 negative (triple-negative), appearing to develop independently of estrogen regulation [72,73]. In non-familial, sporadically occurring ER-alpha-negative breast cancers, decreased expression of BRCA1 protein and reduced ER-alpha mRNA levels reflect a defective interplay between the two regulatory proteins [74]. These findings reveal that *BRCA1* gene mutation inhibits the transcriptional activity of ER-alpha by weakening its liganded activation. In *BRCA1* mutation carriers, impaired estrogen signaling causes increased risk for cancer development, metabolic diseases, and sex hormone imbalance with infertility [46].

In *BRCA1* mutation carriers, decreased estrogen signaling may be the origin of ER-negative and TNBC-type tumors instead of excessive estrogen activation [75]. The significant prevalence of breast cancers among *BRCA1* mutation carrier women may be explained by the unique requirement of the female breast for balanced liganded and unliganded ER activation. In postmenopausal women with type-2 diabetes and obesity, estrogen loss increases metabolic and hormonal imbalances, leading to increased risk of TNBC development.

Molecular investigations have attempted to reveal the key to interactions between ER alpha and BRCA1 proteins. Wild-type BRCA1 protein could suppress the transcriptional activity of ER alpha [76], inhibiting nearly all genes regulated by estrogen [77]. BRCA1 could inhibit p300-mediated ER acetylation, which is crucial for ER transactivation [78]. Conversely, BRCA1 protein could induce upregulation of p300, which is a coactivator of ER alpha [79]. Similarly, BRCA1 could facilitate Cyclin D binding to ER alpha, which upregulates the transcriptional activity of ERs [80]. These ambiguous findings suggest a complex interplay between BRCA1 and ER alpha signaling.

BRCA and ER proteins are also capable of direct binding, regulating each other’s activation [81]. In reality, a decreased expression or inactivation of either ER alpha or BRCA1 protein dysregulates their interaction, jeopardizing both estrogen signaling and genomic stability [46]. Healthy BRCA1 upregulates ER alpha activation via amino–terminus binding; however, its carboxyl–terminus binding has a repressive effect on ER alpha.

In healthy breast cells with *BRCA1* gene mutation, BRCA1 protein synthesis is low, together with decreased expression of ER alpha mRNA and ER alpha protein [82]. In *BRCA1* gene mutation carrier breast cancer cells, the liganded activation of ERs was decreased [75], and ER alpha also showed decreased expression [83].

Molecular investigations have revealed that both healthy and tumor cells carrying *BRCA* mutation are intelligent and respond to the dangers of estrogen defects by driving the regulatory circuits of ERs through different pathways [5]. *BRCA1* mutation promotes an increased expression of epidermal growth factor receptors (EGFs) in healthy epithelial breast cells [84]. In *BRCA* mutation carrier cells, increased growth factor receptor signaling and P13K-Akt cascade activated ERs in an unliganded manner [85]. In fibrous adipocytes of the breast, BRCA protein defects facilitate increased estrogen synthesis by targeting the PII promoter region of the aromatase gene [86]. In mammary luminal progenitor cells with BRCA protein deficiency, a further coactivator of ER alpha, nuclear factor kappa B (NF-κB) showed persistent activation [87] as a compensatory effort to improve estrogen signaling.

In ovarian tumor cells with BRCA protein deficiency, ER alpha showed unusual, conspicuously high unliganded transcriptional activity [88]. In a *BRCA* mutation carrier tumor cell line, increased activation of p300 (an ER coactivator) facilitated the transcriptional activity of ERs [78]. In breast cancer cells with *BRCA* gene mutation, Cyclin D1, another activator of ER transcription, was highly expressed [89].

Mutant variants of p53 (genome-safeguarding protein) were observed in the luminal cells of healthy breasts in *BRCA* mutation carriers [90]. Mutant *TP53* genes were found in both familial and sporadic TNBC-type tumors. These findings support that in germline *BRCA* mutation carriers, somatic mutation of *TP53* may serve as an effort to replace the lost genome-safeguarding function of BRCA protein.

Ovulatory infertility frequently occurs in *BRCA* gene mutation carrier women [91], attributed to impaired liganded estrogen signaling. Early menopause caused by ovarian failure is a key feature in *BRCA* mutation carriers [92]. In most women with *BRCA1* mutation, the lack of functional BRCA1 protein was associated with compensatory increased aromatase levels and abundant estrogen synthesis [93]. In *BRCA* gene mutation carriers, defective ER signaling promotes insulin resistance; however, compensatory high levels of insulin and IGF-1 do not successfully improve glucose uptake [75].

Development of central obesity and metabolic syndrome are markers of insulin resistance and correlate with a higher penetrance of *BRCA* mutation [94]. *BRCA* mutation-associated damage of estrogen signaling disrupts not only genomic stability but also further regulatory circuits of ERs, including those for glucose and lipid metabolism [46].

In conclusion, genomic instability originates from impaired liganded ER activation, rather than from an excessive estrogen concentration. In addition, both healthy and tumor cells attempt to protect against genomic damage by enhancing estrogen signaling via gene amplification and activating mutations. In *BRCA* mutation carrier women, impaired liganded ER activation explains metabolic and fertility disorders and the increased risk of breast cancer.

## 5. Polycystic Ovary Syndrome Originates from the Disruption of Estrogen Signaling via CYP19A Gene Mutation

In premenopausal women with functioning ovaries, polycystic ovary syndrome (PCOS) manifests as a component of insulin resistance syndrome. PCOS presents in adolescent girls with multiple ovarian cysts, menstrual disorders, ovulatory infertility, and hirsutism, in addition to metabolic syndrome and obesity [95]. These findings suggest a failure of genomic estrogen regulation. In young men with insulin resistance, impaired semen quality and infertility may occur in the background of conspicuous metabolic alterations attributed to impaired estrogen signaling [96].

In PCOS, impaired neuro-hormonal signaling along the hypothalamic–pituitary–gonadal axis dysregulates the entire hormonal system [97]. In PCOS cases, cellular glucose uptake is impaired, and insulin resistance leads to compensatory hyperinsulinemia to improve glucose supply to the cells [98]. In vitro studies revealed that high insulin levels may augment both basal and LH-stimulated androgen synthesis in ovarian theca cells, leading to hyperandrogenism in women with PCOS [99].

Development of cystic ovaries is a characteristic symptom in women with serious, genetically defined defects in estrogen signaling, caused by a mutation in the *CYP19* aromatase or *ESR1* gene [100,101]. In aromatase-deficient girls with *CYP19* aromatase gene mutation, cystic ovaries and delayed bone maturation develop during childhood. At puberty, primary amenorrhea, failure of breast development, virilization, and hyper-gonadotrophic hypogonadism may be experienced [100]. In women with PCOS, increased androgen synthesis by the ovaries and adrenal glands occurs as a compensatory response to low estrogen levels; however, conversion from androgen to estrogen is impaired, due to aromatase deficiency [101].

A germline mutation in the *ESR1* gene, which encodes ER alpha, resulted in deep estrogen resistance, delayed puberty, and polycystic ovaries in a young girl [102]. Despite the compensatory sky-high serum levels of estrogen, the absence of breast development, small uterus, and enlarged polycystic ovaries revealed missing estrogen signaling. ER resistance repressed the insulin-assisted uptake of glucose, and hyperinsulinemia was observed as a counteraction to deepening insulin resistance.

Women with PCOS clearly exhibit a higher risk for early cardiovascular diseases, attributed to their metabolic dysfunction and abnormal hormonal pattern [103]. In PCOS cases, increased risk for clotting disorders and low fibrinolytic activity is a characteristic feature that promotes the development of thrombotic complications [104]. In women with PCOS, immune responses are dysregulated. Insulin resistance and hyperandrogenism are associated with immune cell dysfunction and cytokine imbalance, leading to a long-term inflammatory environment [105]. In PCOS cases, all these complications and co-morbidities are associated with impaired aromatase synthesis and decreased estrogen signaling.

Women with PCOS may have an increased risk of developing gynecological tumors. Increased prevalence of endometrial, ovarian, and thyroid cancers was found in PCOS cases [106]. Endocrine disorders with ovulatory infertility, including PCOS, are associated with highly increased risk for endometrial cancer [107].

Recently, PCOS studies in humans and rodents revealed that disruption of estrogen signaling is highly associated with the development of the syndrome [108]. Paradoxically, these findings have led to therapies targeting estrogen, using antiestrogen clomiphene or the aromatase inhibitor letrozole to induce ovulation [109]. Fortunately, the inhibition of earlier defective estrogen signaling provokes extreme counteractions, abruptly upregulating both estrogen synthesis and ER activation, which explains the achievement of good pregnancy outcomes.

## 6. In Type-1 Diabetes, the Characteristic Triad of Insulin Resistance, Fertility Disorder, and Increased Risk for Malignancies Reveals the Impact of Impaired Estrogen Signaling

Type-1 diabetes is an autoimmune condition that leads to the death of pancreatic β-cells, which results in hyperglycemia. The loss of β-cell function progresses gradually and leads to absolute insulin deficiency [110]. Type-1 diabetes patients require life-long insulin replacement therapy. Type-1 diabetes may be diagnosed at nearly any age, but peaks in its presentation occur between ages 5 and 7 and around puberty.

Type-1 diabetes arises in genetically vulnerable patients in whom one or more environmental factors, such as viruses, trigger an autoimmune process, leading to immune-mediated β-cell destruction [111]. To date, little progress has been made in understanding the loss of insulin-producing β-cells in pancreatic islets.

Type-1 diabetes impairs both male and female fertility. In women, type-1 diabetes induces hypogonadism and hyperandrogenism, which can result in decreased fertility [112]. Women with type-1 diabetes have fewer offspring and a higher rate of congenital abnormalities compared with healthy women. Polycystic ovarian syndrome (PCOS) is a common comorbidity in women with type-1 diabetes, with 25–40% of patients showing PCOS development in adolescence [113]. Evidence supports that type-1 diabetes mellitus could impair male fertility, disrupt the gonadal axis, reduce semen quality, and hinder spermatogenesis, due to hyperglycemia and insulin deficiency [114]. Fertility disorders in patients with type-1 diabetes suggest that impaired estrogen signaling underlies both disorders.

Insulin resistance also develops in people with type-1 diabetes mellitus, similarly to those with type-2 diabetes mellitus [115]. Despite improvements in the maintenance of appropriate glucose levels, blood pressure, and the lipid profile, vascular complications (such as coronary artery disease and nephropathy) continue to remain common complications in type-1 diabetes. Risk factors and initiators of insulin resistance in type-1 diabetes appear to be very similar to those in type-2 diabetes [116].

Recent data from studies on humans and rodent models with diabetes suggest that in females, stronger estrogen signaling is protective against β-cell death and type-1 diabetes. Conversely, males are more susceptible to insulin-deficient diabetes due to their higher levels of androgen [117]. In animal studies, ovarian estradiol, used in pharmacological concentrations, protects pancreatic islets against apoptosis [118]. Studies on human and rodent models with aromatase or ER alpha deficiency have challenged the belief that estrogen actions are gender-specific. In addition to the multiple functions of estradiol, it plays crucial roles in the prevention of insulin resistance, diabetes, and obesity in both sexes [117].

Results of animal experiments and human investigations have similarly confirmed that insulin resistance is closely interconnected with the gut microbiome [119,120]. Gut microbiota in type-1 diabetes patients differs in composition and metabolic activity compared with those of healthy subjects and non-autoimmune diabetes models [121]. New evidence suggests that disturbances in the gut microbiome may significantly influence the onset and progression of type-1 diabetes [122]. Since estrogen plays a crucial role in the survival of pancreatic β-cells, the lack of bacteria performing estrogen deconjugation (the estrobolome) may lead to increased danger of apoptotic β-cell death due to decreased free estrogen recirculation.

Diabetes is correlated with hematological malignancies, such as acute lymphocytic leukemia, acute myeloid leukemia, non-Hodgkin lymphoma, and multiple myeloma [123]. Type-1 diabetes in children is strongly associated with an increased risk of acute leukemia, particularly acute lymphoblastic leukemia [124]. In a case–control study in Finland, children with type-1 diabetes showed an increased risk of acute leukemia (OR: 2.0), regardless of whether they were diagnosed before or after the onset of leukemia. Type-1 diabetes and acute lymphoblastic leukemia may share common underlying genetic failures defining disorders of the immune system.

In childhood leukemia, the composition of the gut microbiome significantly influences the outcome of disease in patients undergoing hematopoietic stem cell transplantation. Chemotherapy, antibiotics, and immune suppression lead to dysbiosis of the microbiome, which can lead to serious complications, such as graft-versus-host disease (GVHD) [125]. Specific microbial signatures have been linked to GVHD risk, while interventions like inulin, Lactobacillus, and short-chain fatty acids, particularly butyrate treatment, may help in rectifying the immune functions and improving the outcomes of the disease. Beta-glucuronidase produced by Lactobacilli is a vital regulator of intestinal estrogen metabolism, whereas increased free estrogen levels improve the composition and diversity of gut microbiota [126]. In childhood leukemia, increased intestinal beta-glucuronidase activity improved defensive immune reactions by increasing the free estrogen levels.

Genetic variations of certain alleles of the *IKZF1* gene and its family members are associated with the development of either type-1 diabetes or lymphoid leukemia [127]. Germline *IKZF1* gene mutation predisposes children to developing acute lymphoblastic leukemia [128]. The *IKZF1* gene encodes the transcription factor Ikaros, which regulates the development and function of lymphocytes, thereby affecting the overall immune system activity [129]. The crucial role of the Ikaros transcription factor family in the activation of the immune system highlights that their mutation-associated alterations may highly influence both the autoimmune destruction of pancreatic β-cells and lymphoid leukemia development.

The genetic background of type-1 diabetes has been extensively investigated. Specific variations within the HLA (human leukocyte antigen) region are associated with an increase of about 40–50% in genetic risk for autoimmune damage to pancreatic β-cells [130]. A polygenic risk architecture was established, and a Genetic Risk Score was developed based on 67 causal variants associated with type-1 diabetes for identifying high-risk individuals [131].

Sex modifies genetic risk for type-1 diabetes through interactions with genetic and environmental factors [132]. Epidemiological investigations revealed that males have a higher risk in high-incidence populations. *ESR1* gene mutations do not directly correlate with type-1 diabetes. *ESR1* gene upregulation was observed in the blood of children with type-1 diabetes compared with control cases [133]. This finding suggests that increased expression of the *ESR1* gene may compensate for its mutated partner gene, regulating the function of lymphoid cells. The Pvull CC variant of the *ESR1* gene showed correlations with less inflammatory and angiogenic activity in girls with type-1 diabetes, while the cardiac and metabolic findings were more serious [134]. Considering that inflammation facilitates aromatase synthesis and increases estrogen concentration, this variant of *ESR1* mutation appeared to be independent of the improvement of metabolic status. In type-1 diabetes patients, a single-nucleotide polymorphism of the *CYP19A* gene was associated with decreased risk for cardiovascular complications [135]. In girls with longstanding type-1 diabetes, estrogen receptor alpha polymorphism was associated with improved autoimmune reactions [136].

In conclusion, studies on *ESR1* and *CYP19A* gene mutations in type-1 diabetes cases could not find direct correlations with either metabolic disorders or autoimmune processes.

## 7. Estrogen Is the Principal Regulator of All Cellular Functions in Mammals

Estrogens (estrone, estriol, and estradiol) are unique hormones, as they have no harmfully high concentrations like other hormones do [46]. Activated estrogen receptors (ERs) can choose from several genes for expression and activation. At the same time, they may facilitate or silence the regulatory processes using their coactivators or corepressors. Significant upregulation of estrogen signaling is necessary for ovulation and during pregnancy for the growth of the uterus and embryonic development. Conversely, low levels of estrogen or impaired ER activation are emergency situations, as they lead to genomic dysregulation throughout the body in men and women.

ERs have possibilities for liganded (estrogen-bound) activation through their activation function-2 (AF2) domain and for unliganded activation by transduction molecules, such as growth factor receptors (GFRs) through the ancient activation function-1 (AF1) domain [137]. In estrogen-deficient periods, increased growth factor receptor (GFR) expression and activation may transiently maintain appropriate ER activation through unliganded pathways [138,139]. Balanced liganded and unliganded activation of ERs stimulates estrogen synthesis and ER expression, ensuring DNA stabilization and upregulation of the entire genomic machinery [140]. Artificial inhibition of either liganded or unliganded activation of ERs induces strong compensatory upregulation of the unaffected domain, while the failure of these restorative efforts may lead to downregulation of overall genomic regulation.

The genomic machinery is driven by estradiol (E_2_)-activated ERs by stimulating and silencing all physiological pathways through regulatory circuits [141]. The main regulatory circuits of ERs control DNA stabilization, cell proliferation, and cellular glucose supply. Activated ERs harmonize all physiological actions via balanced upregulation or downregulation of their regulatory circuits. The upregulation of estrogen signaling is always a reparative effort, targeting the restoration of defective genomic functions.

The principal regulatory round of ERs is the *circuit of DNA stabilization* [46]. This process initiates with estrogen activating ER alpha, which then induces upregulation of further ER expression. Abundant activated ERs work by upregulating a genome-safeguarding protein, such as BRCA1. The transcription proteins ER alpha and BRCA1 interact closely via direct binding, enabling them to either enhance or silence each other’s activation. BRCA1 is responsible for the balance between ER alpha expression and aromatase enzyme activation. When ER alpha expression or its activation weakens, BRCA protein facilitates aromatase enzyme expression and activation, leading to compensatory increased estrogen synthesis [142]. In conclusion, the ^E2^ER–BRCA–aromatase–E2–^E2^ER circuit ensures the continuous maintenance of both genome stabilization and estrogen signaling.

In the *regulatory circuit of cell proliferation*, estrogen-activated ERs drive and control cell growth and proliferation at all sites of the body in strong interplay with growth factor receptors, including epidermal growth factor receptors (EGFRs) and insulin-like growth factor-1 receptors (IGF-1Rs) [139]. Estrogen-activated ERs control the expression and activation of growth factors (GFs) and growth factor receptors (GFRs). Transduction of growth factor signal (GFS) induces kinase cascade pathways, sending further unliganded activation to nuclear ERs [51]. Low estrogen levels create an emergency state, leading to increased expression/activation of GFRs and compensatory unliganded activation of ERs. In tumors, therapeutic inhibition of estrogen signaling activates growth factor kinase cascades, providing compensatory stimulation of ER activation through unliganded pathways.

In the *regulatory circuits of glucose supply,* estrogen drives and controls all steps of cellular glucose uptake and glucose homeostasis [143]. Estrogen safeguards the vitality of pancreatic islet cells, protecting them from lipid deposition [144]. It also controls insulin expression, activation, and secretion [145]. In addition, estrogen controls insulin-assisted glucose uptake by supporting the expression and translocation of intracellular glucose transporters (GLUTs) [146]. Defective glucose uptake consistently reflects underlying impaired estrogen signaling [101].

## 8. Adipose Tissue Ensures Metabolic Balance and Energy Homeostasis via Estrogen Regulation

*Circulating estrogen regulates body fat distribution,* remodeling, and the maintenance of adipose tissue health [147]. Males tend to accumulate adipose tissue in visceral locations, promoting metabolic disorders and increasing their risk for chronic conditions, such as cardiovascular diseases and cancers. In contrast, females show inclination to the deposition of subcutaneous fatty tissue in the gluteofemoral region, which is not associated with metabolic disorders. After menopause, estrogen loss in women leads to male-like abdominal deposition of fatty tissue, increasing their susceptibility to metabolic disorders similar to men [148].

Adipose tissue mass is an estrogen-regulated endocrine organ that exhibits the highest activity of estrogen synthesis among extra-gonadal sites [149]. Circulating and locally produced estrogens exert their regulatory effects on adipocyte growth, metabolism, and estrogen signaling in an intracrine manner. In addition, hormonal regulation and energy supply are extended to adjacent organs and tissues in a paracrine manner [150]. Adipose tissue health is primarily regulated by estrogen in collaboration with other hormonal and neural signals [151].

Estrogen exerts its regulatory effects on estrogen-responsive adipocytes, depending on the intensity of their receptor expression [152]. Estrogen signaling regulates glucose metabolism, lipolysis/lipogenesis [48], and energy balance [153,154] in the whole body. In adipose tissue, defects of estrogen signaling result in dysregulation in all regulatory functions of adipocytes, leading to metabolic disorders, obesity, and chronic conditions in fat-regulated organs and tissues [18].

Subcutaneously positioned fatty tissue supplies energy and estrogen regulation for skeletal muscles and the skin. The abdominally located visceral adipose tissue surrounds and protects the visceral organs, gonads, and cardiovascular structures [155]. Female breasts and bone marrow receive abundant energy supply, as breast lobules and bone marrow cell populations are closely integrated with abundant adipose tissue. Female breasts require strict hormonal regulation and high energy supply, synchronized with the hormonal changes of the menstrual cycle [156]. In bone marrow, adipocytes actively influence the proliferation and functional activity of hemopoietic and immune-competent cells via their estrogenic signaling and hormonal secretions [157]. Bone marrow fat also supplies energy and regulatory signals to the bones.

Abdominal adipose tissue has crucial physiological secretory functions [158]. The secretory activities of adipose tissue are defined by its estrogen supply and estrogen sensitivity [147]. Suitable estrogen signaling ensures the health and functional activity of adipocytes. Visceral fat, as a central endocrine organ, regulates metabolic balance, vascular health, appetite, body weight, and the immune system, among many other processes [159].

Sexual hormones, adipokines, cytokines, and growth factors are important signaling molecules in adipose tissue, and their estrogen-regulated activation ensures the health of the whole body. In postmenopausal women and men, most estrogens are produced from circulating adrenal androgens via conversion by the aromatase enzyme [160]. In addition, adipose tissue is capable of locally synthesizing sexual steroids de novo from cholesterol. In adipose tissue, estrogen synthesis and appropriate estrogen signaling regulate the expressions of numerous genes and the harmonized synthesis of various signaling molecules [151].

Adipokines are regulatory hormones secreted by adipose tissue in health and disease. Circulating leptin maintains energy balance in the hypothalamus through anorexinogenic and lipolytic effects. Estrogen increases the expression and activation of leptin receptors, increasing the leptin sensitivity of different cells [161]. Adiponectin improves insulin sensitivity, silencing inflammatory reactions and restoring endothelial functions. In adult mice, oophorectomy facilitated adiponectin synthesis, while estradiol substitution restored normal levels [162]. Resistin is an anti-obesity hormone whose levels increase in parallel with weight gain. Estradiol benzoate treatment decreased resistin levels in subcutaneous adipose tissue [163].

Ghrelin is a multifaceted hormone secreted by gastric and intestinal cells showing endocrine function. Ghrelin is known as the hunger hormone, increasing growth hormone release, food intake, and fat storage when the stomach is empty [164]. Estrogen regulates ghrelin secretion and ghrelin receptor activation, controlling the ghrelin effect on appetite, energy balance, and metabolic changes [165].

In conclusion, estrogen signaling controls all hormones regulating appetite, hunger, food intake, and fat deposition. Impaired ER activation or estrogen deficiency significantly disturbs the balance of food intake and energy expenditure, promoting weight gain and the development of obesity.

Estrogen deficiency and/or ER resistance dysregulates all signaling functions of adipocytes. Adipocytes that lose their ER expression and estrogen-synthesizing capacity exhibit excessive lipid deposition and develop insulin resistance [166]. Considering that impaired estrogen signaling also endangers DNA stability, abdominal obesity is not the origin of cancer development in visceral organs; rather, defects in estrogen function are the common initiators of both obesity and cancer. 

In obese adipocytes, lipogenesis becomes predominant over lipolysis, and deepening estrogen deficiency strengthens this imbalance. In thin stromal adipose cells, increased insulin levels stimulate aromatase expression and estrogen synthesis [167]. This observation supports that in obesity, increased insulin and IGF-1 levels exert anti-obesity effects by facilitating estrogen synthesis. Regulatory defects in obese adipose tissue promote further disorders in the adjacent insulin-resistant organs, resulting in serious co-morbidities, such as metabolic disorders, fatty degeneration, and malignancies [168,169].

In both health and disease, adipose tissue produces inflammation-regulating cytokines that play significant roles in either anti-inflammatory or pro-inflammatory activities. Cytokines are secreted by adipocytes, fibroblasts, and immune competent cells within adipose tissue, influencing systemic metabolic and immune functions [170].

In obesity, adipose tissue mass exhibits low-grade inflammation with abundant macrophages and T-cells [171], targeting improved glucose uptake. Macrophages produce pro-inflammatory cytokines, including tumor necrosis factor alpha (*TNF-α*) and interleukin-6 (IL-6). Pro-inflammatory cytokines drive increased expression and activation of aromatase enzyme, leading to augmented estrogen synthesis [172]. In obese adipose tissue, overexpression of ER alpha increases estrogen sensitivity of fat adipocytes, silences inflammation, rapidly promotes lipolysis, and improves the signaling function [173].

Estrogens have pivotal roles in the improvement of metabolic dysfunctions by orchestrating inflammatory changes and immune responses [159]. Low-grade inflammation in obese, insulin-resistant fatty tissue is a compensatory process to improve the regulatory capacity of adipocytes by increasing aromatase expression and estrogen concentration [5].

A dense inflammatory reaction in the vicinity of HER2-rich and TNBC-type breast tumors is an effort to facilitate aromatase expression and estrogen synthesis, supporting the regulatory improvement of these poorly differentiated cancers [8]. In ER-positive, highly differentiated breast cancers, immune competent cells are not recruited, as adjacent adipocytes increase estrogen synthesis, supporting the genomic repair of tumors. Conversely, tamoxifen blockade of ER-positive tumors activates neighboring inflammation, which is a compensatory response to the sudden inhibition of estrogen signaling.

Adipocytes also synthesize insulin-like growth factor 1 (IGF-1) and exhibit IGF-1 receptor expression (IGF-1R). IGF-1 signaling works within the growth hormone (GH)–IGF–I axis regulating growth, development, and metabolism in mammals, particularly during childhood [174]. GH, produced by the pituitary gland, stimulates the adipose tissue, liver, and skeletal muscles for IGF-1 secretion. IGF-1 then acts directly on various tissues, including bone, muscle, and cartilage, to regulate protein synthesis, cellular growth, and cell division.

IGF-1 regulates glucose and lipid metabolism in collaboration with insulin and estrogen. IGF-1 also regulates glucose and fatty acid uptake, improving insulin sensitivity and reducing body weight, by sharing signaling pathways with insulin [175]. In the early phase of insulin resistance, increased IGF-1 levels coincide with augmented insulin synthesis, leading to compensatory hyperinsulinemia [176]. IGF-1 is an important player in the regulatory circuits of estrogen for controlling cell proliferation and glucose uptake [141].

In adipose tissue, estrogens regulate both IGF-1 synthesis and IGF-1R expression of adipocytes. Estrogen-stimulated IGF-1 synthesis and IGF-1 receptor activation upregulate the AKT and MAPK pathways, facilitating increased unliganded activation of ERs [177]. In the absence of estrogen, IGF-1 receptor signaling maintains the unliganded activation of ERs, ensuring a transient, genome-wide expression of estrogen-regulated genes [137]. In conclusion, in insulin resistance and obesity, upregulation of IGF-1 signaling improves glucose uptake and supports the loss of body weight by increasing the unliganded activation of ERs.

Adipose tissue is a crucial organ in the regulation of immune reactions and immune defense [178]. It actively participates in host defense against pathogens and regulates the quality and intensity of inflammatory processes. In adipose tissue, inflammation serves to improve genomic stability in metabolic diseases and support tumor cells in their self-directed apoptotic death [8].

Estrogen acts as a major regulator of the immune system, influencing the development, function, and activity of various immune cells, including lymphocytes, monocytes, macrophages, and T and B lymphocytes. Estrogen exerts these effects by binding to its specific receptors, which activate gene expression and signaling pathways, modulating both innate and adaptive immunity [179].

Adipocytes and immune competent cells, such as macrophages, T cells, and B cells, interact extensively in both healthy and obese adipose tissue [180]. In lean adipose tissue, eosinophil granulocytes show IL-4 cytokine secretion, and regulatory T cells activate M2-type macrophages, promoting the expression of arginase and anti-inflammatory cytokines, including IL-10. Conversely, in obese adipose tissue, an increased number of M1-type macrophages, and an abundant secretion of pro-inflammatory cytokines, including TNFα and IL-6, may be observed [171]. In obese female mice, estrogen treatment improved insulin sensitivity, silenced inflammation and reduced obesity [181].

In conclusion, insulin resistance and obesity both originate from impaired estrogen signaling. Estrogen loss inhibits glucose uptake and increases the lipid deposition of adipocytes, weakening their regulatory functions. In insulin resistance, hypophysis orchestrates the genome stabilizer processes via neuro-hormonal signaling; however, when worsening trends exceed compensatory efforts, serious co-morbidities develop (Figure 1).

## 9. Skeletal Muscle Contraction Improves Insulin Sensitivity Through Rapid Unliganded Activation of ERs via the IGF-1 Receptor

Physical activity is strongly correlated with human health. Skeletal muscle mass accounts for about 40% of the whole body, and its regulation of glucose metabolism plays a crucial role in maintaining whole-body glucose homeostasis [182]. Regular exercise improves mental health, enhances quality of life, and can improve existing chronic conditions, including obesity and type-2 diabetes [183].

In skeletal muscles, glucose uptake is mainly insulin-dependent at rest, while during exercise, glucose uptake is stimulated by muscle contraction and increased blood flow via insulin-independent pathways [184]. In resting muscle, insulin binding to its membrane receptor triggers a cascade of intracellular signaling, culminating in the activation of GLUT4, which migrates to the plasma membrane and allows glucose entry. In contrast, muscle contraction induces changes in levels of intracellular molecules, such as ATP and calcium, regulating GLUT4 translocation and glucose uptake, even in the absence of insulin.

Estrogen is the principal regulator of cellular glucose uptake and metabolism. Estrogens regulate pancreatic β-cell survival [144] and activate insulin biosynthesis and release [185]. Estradiol treatment facilitates GLUT4 expression and intracellular translocation to the plasma membrane [146]. Impaired estrogen signaling is an emergency state and requires compensatory efforts to improve glucose supply.

Under estrogen regulation, the IGF-1 signaling pathway is strongly linked to insulin signaling, working synergistically on muscle insulin sensitivity, growth, metabolism, and overall health [186]. Insulin and IGF-1 signaling initiate a cascade that activates IRS-1 and IRS-2 signaling proteins, initiating downstream PI3-kinase and MAPK pathways.

Estrogen treatment may stimulate a rapid cellular response within seconds to minutes through the activation of cell membrane-associated ERs [187]. This rapid response avoids the time-consuming transcription and translation processes that can take several hours or even a day. These membrane-located rapid ER responses work through the activation of kinases, calcium mobilization, and G protein cascades, promptly realizing different cellular functions.

The technique of rapid ER response to estrogen was studied on muscle cells and adipocytes [49,146]. Following estradiol treatment, ER alpha promptly translocates from the nucleus to the cell membrane to interact with membrane-associated growth factors and enable rapid, non-genomic signaling. The rapid response to estradiol facilitates GLUT4 expression and translocation to the plasma membrane, improving glucose uptake in both muscle cells and adipocytes.

Facilitating glucose uptake through a rapid response to estrogen may have significant importance in the physiology of mammals, as glucose shortage highly endangers the integrity and function of cells. In addition, rapid activation of non-nuclear estrogen receptor signaling was observed, providing protection against vascular injury [188].

Muscle contraction requires high energy, evoking rapid translocation of ER alpha from the nucleus to the cell membrane, similar to the effect of estrogen treatment. Along the cell membrane, IGF-1 receptors accumulate, waiting to activate unliganded estrogen receptors in an estrogen-deficient environment. ER alpha, arriving at the cell membrane, receives rapid, non-genomic activation from the IGF-1 receptor. Activated ERs induce increased expression and activation of GLUT4 vesicles and promote their migration to and incorporation into the cell membrane of muscle cells. Increased glucose entrance into the muscle cells explains the improvement in insulin resistance attributed to contraction. Membrane-associated ERs activated by IGF-1 receptor also exhibit further possibilities for the activation of nuclear ERs [Figure 2].

Activation of the ER network in the extensive mass of skeletal muscles explains the improvement in glucose uptake throughout the whole body in correlation with physical exercise.

## 10. Estrogen Regulation of Multiple Functions of the Liver

Liver development, integrity, and functional activity are regulated by the neuro-hormonal signaling of hypothalamus–hypophysis system. The GH/IGF-1 axis is one of the major regulators of the liver, which plays a primary role in IGF-1 synthesis [189]. Moreover, estrogen hormones also act on the liver both centrally, controlling pituitary GH secretion, and peripherally, modulating the GHR-JAK2-STAT5 signaling pathway [190]. Estrogens and GHs show thorough interplay in liver physiology, while a disruption of either GH or estrogen signaling leads to dramatic changes in the function of liver both during development and in adulthood.

Estrogen directly regulates the functions of the liver, as it expresses abundant ERα. Estrogen controls liver development, growth, and regeneration [191] and regulates hepatic lipid and glucose metabolism [192].

In addition to the maintenance of hepatic health, estrogen signaling controls multiple functions of the liver, influencing regulation of the whole body. The liver is the main secretor of IGF-1, producing approximately 75%, highly surpassing the IGF-1 synthesis of other tissues, such as adipose tissue and skeletal muscle [193]. Liver-derived IGF-1 behaves as an endocrine hormone, participating in the regulation of further tissues via distribution in the circulation. Conversely, in other tissues, locally secreted IGF-1 may function via paracrine and autocrine mechanisms.

Estrogen controls IGF-1 signaling pathways interacting in a complex manner in the regulation of glucose supply [194]. In turn, the IGF-1 receptor is capable of unliganded activation of ERs via growth-factor kinase cascades. This interplay between ER and the IGF-1 receptor involves multiple levels, including their direct binding. Moreover, ER may modulate IGF-1 binding protein levels, leading to increasing or decreasing bioavailability of IGF-1.

ER and IGF-1 receptor signaling interact in the protection of cardiovascular health. IGF-1 regulates cardiac development and improves the output, stroke volume, contractility, and ejection fraction of the heart [42]. In human studies, IGF-1 stimulates contractility and tissue remodeling, improving heart function after myocardial infarction. Low serum levels of free or total IGF-1 lead to an increased risk of cardiovascular and cerebro-vascular diseases.

Endogenous estrogen is the primary regulator of the liver, ensuring balanced production of clotting and fibrinolytic factors. The liver synthesizes coagulation factors, anticoagulants, proteins involved in fibrinolysis, and thrombopoeitin, a platelet production regulator derived from megakaryocytes [195]. In cirrhotic patients, hepatic dysfunction perturbs the clotting process [196]. The increased risk for thrombosis in patients treated with synthetic estrogen and/or progestin derives from pharmaceutical mistakes [140]. The production of chemically modified hormones, such as ethinylestradiol and medroxyprogesterone, results in a blockade of unliganded ER activation on the AF1 domain and compensatory activation on the AF2 domain. This regulatory imbalance leads to unforeseeable toxic effects of synthetic hormones in human practice.

The liver also plays crucial roles in immune responses [197]. Hepatocytes exhibit immune functions by expressing pattern-recognition receptors (PPRs), secreting complement components and cytokines, and exhibiting immunoglobulin secretion capacity.

In the liver, estrogen controls insulin sensitivity and glucose uptake through balanced activation of glycogen synthesis and glycolysis. In an animal experiment, ER alpha knockout mice showed insulin resistance in both the liver and skeletal muscles [198]. Hepatic lipid metabolism is regulated by estrogens. Estradiol treatment showed antidiabetic and anti-obesity effects in mice kept on a high-fat diet. Estrogen decreased the expression of lipogenic genes in adipose tissue and liver and suppressed the expression of hepatic G-6-Pase. Estradiol treatment in aged rats decreased the lipid peroxidation and improved the functional parameters of the liver [199].

In humans, estrogen plays a pivotal role in the hepatic regulation of serum lipid levels. In postmenopausal women, estrogen loss resulted in higher total and LDL cholesterol levels and increased triglyceride levels compared to premenopausal women [200]. Menopausal estrogen therapy decreases the risk of cardiovascular diseases, likely due to the advantageous changes in plasma lipid profiles [201].

Insulin resistance in hepatocytes dysregulates glucose metabolism, particularly the control of glucose output into the circulation. Hepatic fatty acid synthesis becomes in insulin resistance, leading to hepatic steatosis. Type-2 diabetes is a significant risk factor for progressive hepatic steatosis, from non-alcoholic fatty liver to hepatic cirrhosis [202]. Recent data show that type-2 diabetes and obesity are risk factors for hepatocellular carcinoma [203].

## 11. Hypothalamic Estrogen Signaling Is the Central Regulator of Somatic, Reproductive, and Mental Health

The hypothalamus is situated in the ventral brain above the pituitary gland, creating a central regulatory entity with it [204]. The hypothalamus exhibits an integrative role in linking mental and somatic functions for well-being and adaptability. The hypothalamus continuously receives outer, environmental, and internal stimuli from the body and responds with regulatory commands to maintain the internal balance.

Light signals arriving through the eye play a crucial role in visual perception but also evoke non-image-forming vision by stimulating hypothalamic nuclei. The light–eye–body axis is a neuro-hormonal signaling network that thoroughly influences genomic regulation of the entire body [205]. The hypothalamus regulates hormone production of hypophysis and sends neural impulses through the autonomic nervous system. In the hypothalamus, estrogen is the primary coordinator between the circadian system and gene expression/activation, adapting all mental, somatic, and reproductive functions to changes in light and darkness in mammals [206].

Estrogen is the main regulator of bio-energetic systems in the brain and the body via hypothalamic activation [207]. The hypothalamus–pituitary unit exhibits interplay with sex steroids, modulating all neuro-hormonal signaling deriving from the central nervous system. Estrogen regulates cognitive functions by acting on the hypothalamus and other brain regions, promoting synaptic plasticity and neuron survival through both genomic and non-genomic ER activation [208]. HPG axis-driven regulation shows various connections to stress signaling and their somatic effects [209]. The HPG axis regulates the immune system, showing strong correlations between sex steroid activation and immune responses [210].

Hypothalamic estrogen signaling regulates adipose tissue metabolism and energy homeostasis via the hypothalamus–pituitary–adipose tissue (HPA) axis [211]. Females exhibit gender-specific protection against metabolic diseases, while males exhibit increased disease susceptibility. Adipose tissue regulation is driven by the activation of estrogen receptor alpha (ERα) through neuro-hormonal interactions with other hormones, particularly insulin and adipokines [147].

Hormonal and neural signals from all regions of adipose tissue reach the hypothalamus, conveying information about their own energy homeostasis and those of various adjacent organs and tissues [212]. In certain hypothalamic regions, the loss of ER alpha-positive cells results in obesity and fertility disorders. The hypothalamic arcuate nucleus (ARC) is widely recognized as a central regulator of appetite [213]. The complex hormonal network that regulates appetite, hunger, and food intake is under estrogen control [214].

Disrupted estrogen regulation of the HPA axis contributes to the development of insulin resistance, metabolic syndrome, obesity, and type-2 diabetes, as estrogen plays a pivotal role in the regulation of metabolism and the energy homeostasis of fatty tissue.

## 12. The Origin of Cancer Development from Unhealthy Lifestyle Factors and Bad Habits: Impaired Estrogen Signaling and Associated Insulin Resistance

Much of the burden of cancer may be associated with modifiable lifestyle factors that increase one’s risk for the disease [215]. Epidemiological evidence links major risk factors, such as tobacco use, alcohol consumption, unhealthy dietary patterns, and physical inactivity, to cancer incidence and mortality.

Smoking is the main cause of cancer among harmful lifestyle factors and affects several regions, such as the oral cavity, digestive tract, upper respiratory tract, lung, urinary tract, cervix uteri, and ovaries [216].

Smoking is linked to insulin resistance in a dose-dependent manner [217]. It increases the risk for insulin resistance, mainly via dysregulation of the hormones participating in glucose uptake and causing a shift in lipogenesis and lipid deposition, which promotes abdominal obesity [218]. Studies have demonstrated that in animal models and human tissues, nicotine and its metabolites inhibit aromatase enzyme in a dose-dependent manner, decreasing androstenedione conversion to estrogen [219]. In smokers, nicotine inhibits aromatase enzyme expression and activity in the brain, negatively influencing some basic psychological activities, such as cognition, libido, and appetite [220].

Summarizing smoking-associated insulin resistance and the aromatase inhibition of nicotine, impaired estrogen signaling and resulting glucose uptake failure may explain the strong cancer-inducing capacity of tobacco use at several sites.

Chronic, heavy alcohol consumption dysregulates glucose homeostasis and is associated with the development of insulin resistance [221]. Excessive alcohol consumption is an independent risk factor for type-2 diabetes [222]. Drinking alcohol is closely correlated with cancer development in the oral cavity, pharynx, larynx, esophagus [squamous cell carcinoma only], colorectum, liver [hepatocellular carcinoma only], and female breast [216].

Multiple molecular mechanisms contribute to cancer development mediated by drinking [223]. Acetaldehyde, deriving from ethanol metabolism, is a potent carcinogen capable of inducing DNA damage and mutations. During alcohol metabolism, oxidative stress can damage cellular components, including DNA, proteins, and lipids, contributing to genetic mutations and genomic instability [224].

Alcohol abuse disrupts the hypothalamic regulation of the endocrine system, affecting various hormones that regulate growth, metabolism, stress, and reproduction, leading to metabolic disorders, cognitive decline, and infertility [225]. Alcohol upregulates aromatase activity and increases estrogen levels in women who drink heavily, which are presumably the causal factors for their increased risk for breast and ovarian cancers [226]. In reality, the increased aromatase activity and estrogen synthesis in drinkers are compensatory responses to the inhibition of ERs and developing insulin resistance.

The toxic effects of ethanol were studied on osteoblasts in female rats and on cultured osteoblasts in vitro [227]. Ethanol inhibited estrogen receptor signaling by blocking both ER alpha and ER beta activation, while estradiol treatment restored the health of osteoblasts. These results explain that bone loss develops in highly alcohol-exposed females, while estrogen treatment exhibits protective effects.

Alcohol-induced blockade of ER activation dysregulates all genomic functions and increases the risk for several cancers in both men and women, despite the increased compensatory estrogen concentration. The aromatase inhibition from smoking and the ER blockade from drinking clearly explain the amplification of cancer risk in patients who have both smoking and drinking habits.

Unhealthy dietary patterns have great role in the development of obesity and increased cancer risk. Overeating and excess body weights are associated with increased risk for cancer in the esophagus (adenocarcinoma only), gastric cardia, colorectum, liver, pancreas, endometrium, ovary, kidney, thyroid, and female breast (postmenopausal cancers only) [228]. Interestingly, among premenopausal women, consistent inverse associations have been observed between obesity and breast cancer risk [229]. Overeating in premenopausal women with healthy hormonal balance leads to gluteofemoral adiposity, which is independent of metabolic disorders. It is not obesity but rather cycling estrogen levels and the insulin sensitivity of abdominal adipocytes that protects obese young women from metabolic diseases and breast cancer [230].

Diet quality is a significant factor in the development of insulin resistance. Studies show that a high-energy, high-fat, high-carbohydrate, and low-fiber diet, as well as eating ultra-processed foods, can increase risk for type-2 diabetes [231]. Diet and nutrition appear to be modifiable risk factors for the development of several cancers, but their associations may be faulty due to the inherent bias of investigations [232]. Only a few single foods or nutrients appeared to be strongly or suggestively associated with cancer risk, as tumor development is influenced by multiple genetic and environmental factors beyond diet quality.

Sedentary lifestyle has wide-ranging adverse impacts on the human body, including high risks of metabolic disorders, such as diabetes mellitus and obesity, musculoskeletal disorders such as sarcopenia and osteoporosis, as well as depression and cognitive impairment [233]. Lack of physical activity increases the risk for cardiovascular diseases and cancers and raises all-cause mortality across populations.

Physical activity is strongly linked to a lower risk of developing several types of cancer, including bladder, breast, colon, and lung cancer [234]. Regular exercise decreases cancer risk by maintaining a healthy weight, regulating hormonal balance, particularly that between estrogen and insulin, and strengthening the immune system. Experimental studies showed that physical activity improved insulin sensitivity through skeletal muscle contraction both immediately during exercise and for up to 48 h afterward [235]. Muscle contraction can increase glucose uptake through coordinated increases in microvascular perfusion and glycogen synthase activity.

Considering the crucial role of estrogen signaling in glucose uptake, muscle contraction may be associated with rapid, non-genomic ER activation, facilitating glucose uptake in the absence of insulin.

## 13. Discussion

Tumorigenesis is a multistep process, initiating with oncogenic mutations in a normal cell. The next steps are the pervasive somatic mutations conferring clonal advantage in the neighborhood. Pervasive somatic mutations and clonal expansion in tumor-free tissues rarely lead to transformation into cancer [236]. Recently, researchers have suggested that environmental cancer risk factors may highly influence the early clonal expansion and malignant transformation of mutated cells [237].

Recent research strongly suggests that environmental and lifestyle factors are the primary drivers of cancer. About 93% of all cancers are nonhereditary, highlighting the significance of environmental and lifestyle factors in carcinogenesis. Conversely, inherited genetic defects are linked to only about 5–6% of cancer cases [238].

In cases with germline mutation of *BRCA1/2* genes, all somatic cells are affected by the genetic failure. In *BRCA* mutation carrier women, breast and ovarian cancer risks are highly increased, although well-working BRCA proteins have crucial roles in the genomic stability of all cell types [69,70].

BRCA1 protein is a crucial member of the genome stabilizer circuit ensuring the equilibrium between liganded ER activation and the estrogen synthesis of aromatase enzyme [142]. Mutation-associated alteration in BRCA1 protein inhibits the liganded activation of ERs, while a compensatory increased unliganded ER activation was proven in the affected cells [76]. The genetically defined ER resistance is a stress at the cellular level inspiring both local and hypothalamic counteractions for the improvement of estrogen signaling and genomic stability. Cells with germline *BRCA* mutations increase their estrogen synthesis [86] and activate the gene of p53 protein via somatic mutation [90]. The hypothalamus improves estrogen synthesis in the gonads, and stimulates growth factor secretion via the GH-IGF-1 axis, promoting the unliganded activation of ERs in the regulatory circuits of cell proliferation and glucose uptake [141].

Among *BRCA* gene mutation carriers, breast and ovarian cancer risk is highly increased in spite of all compensatory mechanisms as these organs show extreme necessity for estrogen-controlled regulation. Examining the influences of lifestyle factors, low body weight and moderate physical activity moderately decreased breast cancer risk in premenopausal women with *BRCA* gene mutation, while did not affect the risk of postmenopausal women [239]. In *BRCA* mutation carriers, smoking further increased the risk for breast cancer, while alcohol consumption could not significantly modify breast cancer risk.

In conclusion, germline mutations of *BRCA* genes affect all somatic cells; consequently, all are carrying oncogenic mutations. They fight against genomic instability via protective somatic mutations and they can achieve a delay of cancer development, or in the minority of cases, cancer does not develop. Germline oncogenic mutations are controversial to the theory of multistep carcinogenesis. Oncogenic mutation in a single cell may be a frequent event in the human body. However, the intelligent cell can fight against its genomic instability instead of generating invasive colonies with oncogenic mutations.

Studies on insulin resistance, including type-1 and type-2 diabetes, provided great possibilities for cancer research as insulin resistance and its comorbidities are strongly associated with increased cancer risk. These two types of diabetes apparently seem to be highly different in their origin, while their progression and the developing co-morbidities show quite similar features.

Defect of estrogen signaling seems to be a common origin of both types of diabetes as estrogen-regulated genes control the metabolic and hormonal equilibrium as well as immune cell development and function [141]. Either in type-1 or type-2 diabetes, a characteristic triad of impaired estrogen signaling may be experienced: fertility disorder, insulin resistance and increased cancer risk.

The origin of insulin resistance may be associated with genetically defined familiar inclination. Nevertheless, in the majority of cases lifestyle factors seem to be linked with developing metabolic and hormonal disorders. The vast majority of genetic loci associated with type-2 diabetes development are primarily linked with pancreatic β-cell development and function as well as insulin secretion [240]. Genes associated with cellular glucose transport are rarely affected. In type-2 diabetes cases, the circuit of *ESR1*, *BRCA* and *CYP19* genes does not show mutation and the upregulation of estrogen signaling may easily compensate the difficulties of glucose uptake circuit. Since the development of type-2 diabetes is associated with ageing and menopause in women, the genetic inclination may be manifested in correlation with weakening compensatory efforts. In cases without genetic alterations, lifestyle factors and bad habits may promote the development of insulin resistance. Overeating, excessive sugar consumption, lack of physical activity, smoking and alcohol drinking downregulate the main genomic circuit and insulin resistance develops.

Insulin resistance has an insidious onset and through hyperinsulinemia and metabolic syndrome type-2 diabetes develops. In hyperinsulinemia, the increased insulin secretion of pancreatic β-cells can recruit further partners for the facilitation of glucose uptake and the maintenance of normal glucose level. In metabolic syndrome, several molecular changes can be observed that worsen the metabolic and hormonal disorders, while others have improving efforts for the facilitation of aromatase expression and ER reactivation. Increased insulin and IGF-1 signaling stimulate inflammatory reactions in insulin-resistant adipose tissue [51]. Increased expression of proinflammatory cytokines stimulates aromatase expression and increases estrogen concentration [52]. Increased estrogen concentration restores glucose uptake and a decreasing estrogen level silences the inflammation [8]. Increased expression of IGF-1 alone has a direct role in glucose uptake via unliganded ER activation [50]. Insulin and estrogen signaling together facilitate intracellular glucose transport [49]. In conclusion, the origin of insulin resistance is a downregulation of estrogen-controlled genes. Restoration of estrogen signaling improves glucose uptake and metabolic homeostasis [47].

Numerous epidemiological studies and randomized trials supported that lifestyle modification associated glycemic benefits may prevent type-2 diabetes. Moreover, lifestyle improvement may reduce the development of type-2 diabetes, the associated comorbidities and mortality [241].

The onset of type-1 diabetes shows an autoimmune inflammatory process destructing pancreatic β-cells and leading to a lifelong necessity of insulin substitution for patients [110]. In patients with type-1 diabetes, additional autoimmune diseases may develop, most frequently Hashimoto’s thyroiditis in 17–30% of cases [242].

The role of estrogens in the development of autoimmune processes seems to be controversial. The reasons why women have stronger immune systems, but higher incidence of autoimmunity, are not clarified [243]. Considering that estrogens are the chief regulators of immune responses [179], autoimmunity is a failure of the immune system attributed to impaired estrogen signaling. Recently, estrogen treatment proved to be protective against experimental autoimmune disease by regulation of cytokine expression and Th2 cell activity [244].

Recent data suggest that specific dysbiosis of the gut microbiome may be associated with the onset and progression of type-1 diabetes [122]. Intestinal Lactobacilli with beta glucuronidase activity perform estrogen deconjugation and support β-cell survival via allowing the recirculation of free estrogens [62]. The possibility of prevention and treatment of type-1 diabetes arises by appropriate microbiome treatment.

## 14. Conclusions

Reduced estrogen signaling caused by either endogenous or environmental factors is alarming for the hypothalamus, which sends neural and hormonal signals throughout the whole body to restore genomic stability.

In the first, compensated phase of defective estrogen signaling, patients appear healthy. Hyperinsulinemia is a subtle signal of the genome indicating successful restoration of estrogen signaling, but it occurs at the expense of excessive insulin synthesis. Over time, impaired estrogen signaling and glucose uptake result in metabolic syndrome, characterized by various symptoms and altered laboratory findings. Compensatory molecular changes are warnings that the genome is struggling to restore estrogen signaling and DNA stability. Increasingly impaired estrogen signaling leads to type-2 diabetes, showing increased serum glucose levels despite all compensatory actions. Simultaneously, organs suffer from the lack of glucose and excessive lipid deposition. The complete breakdown of estrogen signaling results in DNA damage and cancer initiation in the affected organ. Cancer development is the genome’s cry for help to combat against dysregulation.

## Figures and Tables

**Figure 1 cancers-18-00078-f001:**
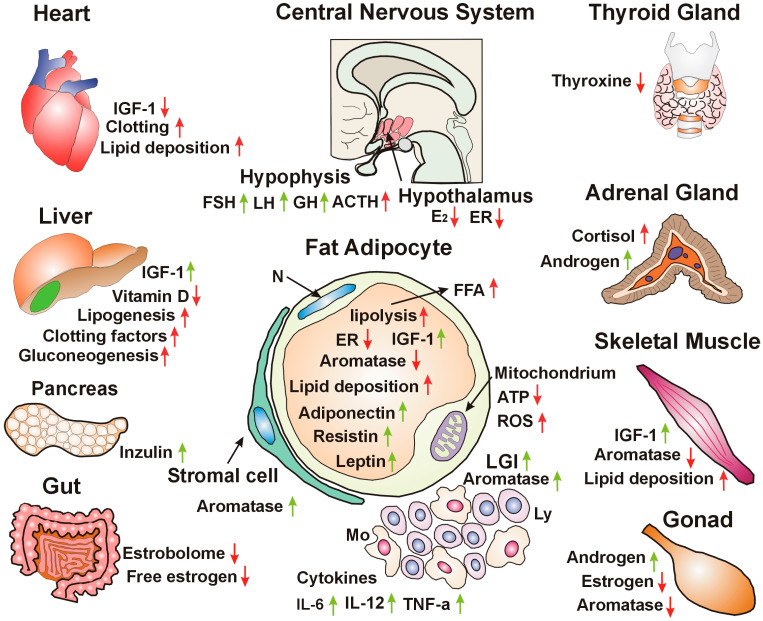
Hypothalamic nuclei detect reduced estrogen (E_2_) concentration and the decreased expression of estrogen receptors (ERs), deteriorating estrogen signaling and cellular glucose uptake. Hypophysis increases follicle-stimulating hormone (FSH) and luteinizing hormone (LH), stimulating the gonads to produce sex steroids. Increased growth hormone (GH) secretion facilitates IGF-1 production in different organs, improving hormonal regulation. Adrenocorticotropic hormone (ACTH) stimulates the adrenal gland to produce sex steroids and cortisol. The heart experiences lipid deposition, decreased insulin-like growth factor 1 (IGF-1) levels, and dysregulation of clotting factors. The liver shows compensatory increased IGF-1 synthesis, decreased vitamin D synthesis, and increased lipogenesis, gluconeogenesis, and coagulation factor synthesis, further disrupting genomic regulation. Pancreatic β-cells exhibit compensatory increased insulin synthesis. The gut loses the bacteria of the estrobolome that are crucial for the reactivation of bound estrogens and allow the recirculation of free estrogens. Fat adipocytes show decreased estrogen receptor (ER) and aromatase expression, but increased compensatory IGF-1 synthesis. Increased lipolysis leads to high free fatty acid (FFA) levels and lipid deposition, even in non-adipose tissues. Stromal adipose cells are capable of increased aromatase synthesis. Mitochondria show functional disorders, leading to decreased adenosine triphosphate (ATP) levels, while an increased production of reactive oxygen species (ROS) shows their oxidative stress. Low-grade inflammation (LGI), comprising monocytes and lymphocytes represents an immune reaction. Pro-inflammatory cytokines (IL-6, IL-12, TNFα) increase aromatase expression and estrogen concentration. The thyroid gland shows lower thyroxine synthesis, worsening genomic functions. Adrenal gland stress increases cortisol synthesis, further worsening glucose tolerance. Androgen level increases are attributed to aromatase deficiency. Skeletal muscle shows compensatory increased IGF-1 signaling, while decreased aromatase and estrogen synthesis increase lipid deposition. Gonads are stimulated for increased androgen synthesis, while the lack of aromatase activity leads to estrogen deficiency. N: nucleus. Red arrows: worsening insulin resistance. Green arrows: efforts for improving estrogen signaling.

**Figure 2 cancers-18-00078-f002:**
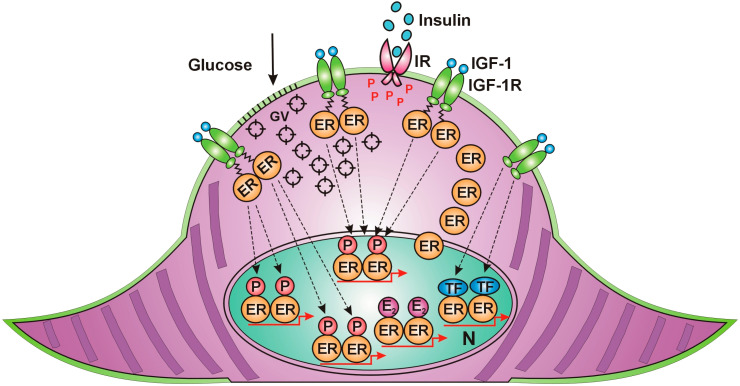
Skeletal muscle cell in contraction. (1) Estrogen receptors (ERs) migrate from the nucleus to the cell membrane. (2) Insulin-like growth factor 1 (IGF-1)-activated receptors (IGF-1Rs) carry out rapid, unliganded activation on the arriving ERs. (3) Activated ERs induce GLUT4 expression and rapid translocation of GLUT4 vesicles (GVs) to the cell membrane. (4) The incorporation of GLUT4s into the cell membrane allows glucose entry into the cytoplasm. (5) Membrane-associated, IGF-1R-activated ERs may transfer their activation to nuclear ERs. IR: insulin receptor, P: phosphorylation, E_2_. estradiol, TF: transcription factor, N: nucleus.

**Table 1 cancers-18-00078-t001:** Opinions on the origin of insulin resistance and the role of hyperinsulinemia.

Theories	References
Insulin resistance is the primary alteration	Wilcox 2005 [30]
Insulin resistance has various causal factors	Johnson et al. 2013 [31]
Low-grade inflammation is the origin of insulin resistance	Khodabandehloo et al. 2016 [32]
Lipotoxicity may impair glucose uptake	Elkanawati et al. 2024 [33]
Altered gut microbiome causes insulin resistance	Jang et al. 2021 [34]
Mitochondrial oxidative stress causes insulin resistance	Fazakerley et al. 2018 [35]
Impaired GLUT4 trafficking causes insulin resistance	van Gerven et al. 2023 [36]
Hyperinsulinemia is the primary alteration attributed to energy rich diet	Freeman et al. 2025 [37] Janssen 2025 [38] Zhang et al. 2021 [39]
Interplay among insulin, GH and IGF-I leads to hyperinsulinemia and insulin resistance	Nijenhuis-Noort et al. 2024 [40]
Hyperinsulinemia improves insulin resistance via activation of IGF-1 signaling	Giustina et al. 2015 [41] Macvanin et al. 2023 [42]

**Table 2 cancers-18-00078-t002:** Downregulation of estrogen-controlled genes is the origin of insulin resistance.

Findings	References
Estrogen controls genome stabilization, cell proliferation, and glucose supply	Suba 2015 [46]
Restoration of estrogen signaling improves glucose uptake and metabolic homeostasis	Yan et al. 2019 [47] Kurylowitz 2023 [48]
Insulin and estrogen signaling work together to regulate GLUT4 transport	Gregorio et al. 2021 [49]
Insulin-like growth factor 1(IGF-1) has a direct effect on glucose uptake both with and without insulin	Rajpathak et al. 2009 [50]
IGF-1 signaling is capable of the unliganded activation of estrogen receptors even in the absence of estrogen	Suba 2020 [51]
In insulin resistance, simultaneously increased insulin and IGF-1 signaling stimulate inflammation	Salminen et al. 2021 [52]
In insulin resistance, increasing expression of pro-inflammatory cytokines stimulates aromatase expression and estrogen concentration	Ohlsson et al. 2017 [53]
In insulin resistance, increased estrogen levels restore glucose uptake and a decreasing estrogen concentration reduces inflammation	Suba 2024 [8]
Estrogen and insulin simultaneously regulate the balance between lipolysis and lipogenesis. In insulin resistance, impaired estrogen function liberates free fatty acids.	Saponaro et al. 2015 [54]
In obesity, abundant free fatty acids in the circulation are pathologically deposited in non-adipose organs, a process attributed to weak estrogen signaling.	Chakrabarti et al. 2013 [55] Ro et al. 2020 [56]
The estrobolome, gut bacteria with β-glucuronidase activity, reactivates bound estrogens, improves insulin sensitivity, and supports the survival of pancreatic β-cells.	Wang et al. 2025 [57]
Insulin and estrogen signaling regulate autophagy and metabolism in mitochondria in close partnership. In insulin resistance, weak estrogen signaling leads to impaired mitochondrial function.	Tao et al. 2023 [58]

## Data Availability

No new data were created for this work.

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
