# Peer review of "Human Cancers Derived from Either Genetic or Lifestyle Factors Are Initiated by Impaired Estrogen Signaling"

_cancers, 2025, doi:10.3390/cancers18010078_

Round 1
Reviewer 1 Report
Comments and Suggestions for Authors
The review “Human Cancers Deriving from Either Genetic or Lifestyle Factors are Initiated by the Damage of Estrogen Signaling” discusses modern aspects of human cancer. Suggestions:
- The introduction is too long and contains excessive information that does not add value to the subject.
- Chapter 2 is interesting but requires more studies from the literature and further development of the topic.
- Add tables in each subchapter to summarize the articles presented from the literature and to highlight the conclusions related to each pathology.
- Include an additional figure illustrating the pathophysiology discussed in Chapter 7.
- Figure 1 needs to be professionally redesigned.
- Add a new chapter on leukemia. Discuss the importance of the microbiota as a central element, especially in blood cancer pathologies – recommended reference: 10.3390/children12020166.
- Chapter 11 should be removed.
- Add a Discussion chapter presenting areas in the literature that require further studies.
- The conclusions are too long and should be limited to one paragraph.
Author Response
Comments and Suggestions of Reviewer
1. The introduction is too long and contains excessive information that does not add value to the subject.
2. Chapter 2. is interesting but requires more studies from the literature and further development of the topic.
- Add tables in each subchapter to summarize the articles presented from the literature and to highlight the conclusions related to each pathology.
- Include an additional figure illustrating the pathophysiology discussed in Chapter 7.
- Figure 1 needs to be professionally redesigned.
- Add a new chapter on leukemia. Discuss the importance of the microbiota as a central element, especially in blood pathologies – recommended reference: 10.3390/children12020166.
- Chapter 11. should be removed.
- Add a Discussion chapter presenting areas in the literature that require further studies.
- The conclusions are too long and should be limited to one paragraph.
Responses to Comments and Suggestions of Reviewer
1.The introduction is too long and contains excessive information that does not add value to the subject.
Answer: The introduction vas shortened.
2.Chapter 2. is interesting but requires more studies from the literature and further development of the topic.
Answer: Further studies were included.
- Add tables in each subchapter to summarize the articles presented from the literature and to highlight the conclusions related to each pathology.
Answer: Tables were added to the 2nd and 3rd chapters.
- Include an additional figure illustrating the pathophysiology discussed in Chapter 7.
Answer: Additional figure 1. was included. (now Chapter 8).
- Figure 1 needs to be professionally redesigned.
Answer: Earlier Figure 1 was rejected.
- Add a new chapter on leukemia. Discuss the importance of the microbiota as a central element, especially in blood pathologies – recommended reference:
10.3390/children12020166.
Answer: A new chapter was included on Type-1 diabetes and its correlations with childhood leukemia. The importance of microbiota in both diseases was discussed (Chapter 6.)
- Chapter 11. should be removed.
Answer: Lifestyle factors and bad habits strongly affect estrogen and insulin signaling in correlation with cancer development. This chapter shows that there is no need of cancer initiator oncogene, lifestyle may weaken estrogen signaling and increases cancer risk.
- Add a Discussion chapter presenting areas in the literature that require further studies.
Answer: Discussion chapter was rewritten.
- The conclusions are too long and should be limited to one paragraph.
Answer: Conclusion was shortened.
Reviewer 2 Report
Comments and Suggestions for Authors
This review attempts to link the dysregulation of estrogen signaling—whether by genetic, epigenetic, or environmental insults—to the initiation and progression of multiple human cancers. The authors argue that damage to estrogenic homeostasis serves as a unifying oncogenic mechanism underlying diverse malignancies. While the conceptual framing is ambitious, the manuscript suffers from severe structural, scientific, and stylistic shortcomings. It conflates correlation with causation, lacks mechanistic precision, and reads more as a speculative essay than a rigorously researched scientific review. In its current form, it fails to meet the standards of Cancers or comparable oncology journals.
Comments
- The central premise, that all human cancers, regardless of etiology, originate from disrupted estrogen signaling, is biologically implausible and unsupported by evidence. While estrogen plays an established role in hormone-dependent cancers (e.g., breast, ovarian, endometrial), extrapolating this to all tumor types (e.g., pancreatic, hematologic, glioma) is conceptually unsound and contradicts well-established oncogenic pathways.
- The paper exhibits serious overgeneralization. Terms like “damage of estrogen signaling” are repeatedly used without definition. Does this refer to receptor mutation, hormonal imbalance, metabolic interference, or environmental endocrine disruption? The lack of operational clarity renders the argument vague and untestable.
- The review lacks methodological structure. There is no systematic literature retrieval or inclusion criteria. The text reads as a collection of loosely connected assertions rather than a coherent synthesis of peer-reviewed evidence. A modern scientific review—especially in Cancers—requires transparent selection and critical evaluation of sources, not anecdotal citation chains.
- The molecular discussion is superficial and often inaccurate. The authors mention ERα and ERβ but fail to distinguish their opposing roles in proliferation and differentiation. There is no mention of non-genomic estrogen signaling via GPER1 or of context-dependent receptor crosstalk (e.g., with PI3K/AKT or MAPK). Furthermore, key regulatory co-factors such as SRC-1, NCOA, and FOXA1 are absent, suggesting a limited grasp of estrogen receptor biology.
- The proposed mechanism of “estrogen signaling damage leading to all cancers” ignores tumor heterogeneity and the multistep nature of carcinogenesis. Many cancers—such as lung adenocarcinoma, melanoma, and glioblastoma—arise from mutations in non-hormonal pathways (e.g., KRAS, BRAF, EGFR, TP53) with no estrogenic mediation. The manuscript never reconciles these contradictions.
- The role of environmental and lifestyle factors is presented simplistically. Statements such as “pollution and diet destroy estrogen signaling, leading to cancer” are speculative and lack mechanistic substantiation. The discussion does not reference relevant epidemiological evidence, nor does it differentiate endocrine disruptors (e.g., BPA, dioxins) from other carcinogenic exposures (e.g., tobacco, UV radiation).
- The article conflates association with causation throughout. The authors cite correlations between estrogen metabolites and cancer incidence as proof of mechanistic causality, ignoring confounding variables such as age, adiposity, or metabolic comorbidities.
- Figures (if any) are schematic and overly simplistic. They fail to represent molecular details such as receptor dimerization, transcriptional coactivation, or pathway cross-talk. A robust mechanistic review would require well-annotated figures linking estrogen signaling to DNA damage, inflammation, and tumor microenvironment remodeling.
- The tone of writing is unscientific and occasionally polemical. Phrases such as “all cancers arise from estrogen damage” and “modern lifestyle breaks hormonal harmony” read more like advocacy statements than scientific reasoning. The manuscript would benefit from restraint, precision, and evidence-based argumentation.
- The discussion ignores well-established counterexamples and contradictions. For instance, estrogen exerts protective effects in certain tissues (e.g., colon, heart, brain). The authors’ one-directional framing of estrogen as uniformly oncogenic is inaccurate and fails to reflect the bidirectional nature of its biological roles.
- References are poorly curated and imbalanced. Key studies in estrogen receptor structure–function, endocrine disruption mechanisms, and hormone-related carcinogenesis are omitted. Many citations are outdated, secondary sources, or lack DOI verification.
- The manuscript has no coherent conclusion or future perspective. A credible review should identify gaps in knowledge and propose research directions—e.g., how estrogen signaling interacts with metabolic or immune pathways in specific cancer types—rather than making sweeping universal claims.
- The English is understandable but stylistically inconsistent, with multiple grammatical errors, imprecise phrasing, and poor paragraph transitions. Substantial language and logic editing would be necessary even after conceptual revision.
Author Response
RESPONSE TO Reviewer 2.
Open Review
Comments and Suggestions for Authors
This review attempts to link the dysregulation of estrogen signaling—whether by genetic, epigenetic, or environmental insults—to the initiation and progression of multiple human cancers. The authors argue that damage to estrogenic homeostasis serves as a unifying oncogenic mechanism underlying diverse malignancies. While the conceptual framing is ambitious, the manuscript suffers from severe structural, scientific, and stylistic shortcomings. It conflates correlation with causation, lacks mechanistic precision, and reads more as a speculative essay than a rigorously researched scientific review. In its current form, it fails to meet the standards of Cancers or comparable oncology journals.
ANSWER: Completion: Insulin resistance, the well-known cancer risk factor was originated from the dysregulation of estrogen signaling. Estrogen signaling instead of “homeostasis”.
Comments
- The central premise, that all human cancers, regardless of etiology, originate from disrupted estrogen signaling, is biologically implausible and unsupported by evidence. While estrogen plays an established role in hormone-dependent cancers (e.g., breast, ovarian, endometrial), extrapolating this to all tumor types (e.g., pancreatic, hematologic, glioma) is conceptually unsound and contradicts well-established oncogenic pathways.
ANSWER: There are numerous epidemiological, experimental and genetic studies justifying the principal regulatory capacities and anticancer effects of ovarian estrogens in several organs. In this work, the principle of “hormone dependent cancers” was not analyzed and was not extended to all tumors, this is a misunderstanding. Insulin resistance (IR) is a well-known cancer risk factor in several regions, and this work established that a defect of estrogen signaling is the origin of insulin resistance instead of overeating or hyperinsulinemia. The “bipolar” effect of estrogen was mistakenly established on postmenopausal patients underwent to synthetic estrogens and synthetic progestins used as MHT and contraception. Modified synthetic hormones are inhibitors of ERs and cause insulin resistance and its comorbidities including breast cancer [Ref.140 and 141]. This work does not analyze the toxic effects of synthetic hormones and their unforeseeable risk for breast cancer.
- The paper exhibits serious overgeneralization. Terms like “damage of estrogen signaling” are repeatedly used without definition. Does this refer to receptor mutation, hormonal imbalance, metabolic interference, or environmental endocrine disruption? The lack of operational clarity renders the argument vague and untestable.
ANSWER: In 4th and 5th chapters, molecular studies support that in genetically defined increased cancer risk, either ER resistance or aromatase deficiency maybe the causal factor of defective estrogen signaling, insulin resistance and increased cancer risk, particularly in female organs. The mentioned “hormonal imbalance” and “metabolic interference” (this is not my phrase) are consequences of the defect of estrogenic regulation. Environmental endocrine disruption by xenoestrogens is not included into this study. There is no overgeneralization. Insulin resistance generalized itself among several organs.
- The review lacks methodological structure. There is no systematic literature retrieval or inclusion criteria. The text reads as a collection of loosely connected assertions rather than a coherent synthesis of peer-reviewed evidence. A modern scientific review—especially in Cancers—requires transparent selection and critical evaluation of sources, not anecdotal citation chains.
ANSWER: Literary data are voluminous and highly controversial regarding both estrogen- induced cancer and the mechanism of cancer development in insulin resistance. Systematic literature retrieval or inclusion criteria are irrational in the cavalcade of ambiguous theories. The origin of insulin resistance is highly debated and unknown even today. Providing a coherent synthesis, peer-reviewed evidences were accumulated justifying the correlation between insulin resistance and the defect of estrogen signaling [Chapter 3].
- The molecular discussion is superficial and often inaccurate. The authors mention ERα and ERβ but fail to distinguish their opposing roles in proliferation and differentiation. There is no mention of non-genomic estrogen signaling via GPER1 or of context-dependent receptor crosstalk (e.g., with PI3K/AKT or MAPK). Furthermore, key regulatory co-factors such as SRC-1, NCOA, and FOXA1 are absent, suggesting a limited grasp of estrogen receptor biology.
ANSWER: This work had no intention to describe the multitude mechanisms of activated ER signaling and to describe its numerous co-activators and co-repressors, it would require a voluminous article. Excellent basic works were cited [refs 137, 139, 143, 146]. Unliganded activation of membrane associated ERs by growth factors is a crucial mechanism in estrogen-deficient periods. The significance of PI3K/AKT and MAPK regulatory pathways in unliganded ER activation was written in Chapter 8, on p. 17. and Chapter 9, on p.19.
- The proposed mechanism of “estrogen signaling damage leading to all cancers” ignores tumor heterogeneity and the multistep nature of carcinogenesis. Many cancers—such as lung adenocarcinoma, melanoma, and glioblastoma—arise from mutations in non-hormonal pathways (e.g., KRAS, BRAF, EGFR, TP53) with no estrogenic mediation. The manuscript never reconciles these contradictions.
ANSWER: Oncogenic mutation and a multistep nature of carcinogenesis is debated, as somatic mutations of numerous genes occur in cancer free patients as well, and they may ensure even an increased fitness for affected organs [ref. 5]. This argument will be included into Discussion (chapter 13) in red. In addition, the occurrence of KRAS, BRAF, EGFR, TP53 somatic gene mutations in tumors, all are associated with efforts for improving ER signaling, since estrogen controls all immune processes, growth factors, genome safeguarding proteins and many other players of regulation. My article dealing with somatic mutations, as efforts for DNA stabilization (ref. 5) gained a Distinguished Cancer Treatment Innovation Award from Scifax Company (India), in 2024, following invited application.
- The role of environmental and lifestyle factors is presented simplistically. Statements such as “pollution and diet destroy estrogen signaling, leading to cancer” are speculative and lack mechanistic substantiation. The discussion does not reference relevant epidemiological evidence, nor does it differentiate endocrine disruptors (e.g., BPA, dioxins) from other carcinogenic exposures (e.g., tobacco, UV radiation).
ANSWER: The section 12 was concentrated only to lifestyle factors, according to the suggested topic of the editor of special issue. Harmful environmental factors (pollution, radiation, endocrine disruptors) were not included. Lifestyle factors, such as smoking and alcohol consumption are in close correlation with aromatase inhibition (refs. 219, 220) and ER blockade (ref. 227), which strongly support their increased cancer risk. Overeating and refined sugar consumption cause alterations in the gut microbiome leading to insulin resistance (ref. 58). This observation supports that the lack of estrobolome and a decreased recirculation of free estrogens leads to IR and the associated increased cancer risk. Sedentary lifestyle is a well-known risk for cancer. Skeletal muscle contraction accelerates ER migration from the nucleus to the membrane and IGF-1 receptor rapidly activates ERs without estrogen. Rapid ER activation leads to GLUT4 translocation and improved glucose uptake (ref. 146). These data are not “simplistic speculations” but rather, here is the first, exact explanation of correlations between muscle contraction and improved glucose uptake. At the same time, it reflects the pivotal role of ER signaling in the improvement of insulin resistance
- The article conflates association with causation throughout. The authors cite correlations between estrogen metabolites and cancer incidence as proof of mechanistic causality, ignoring confounding variables such as age, adiposity, or metabolic comorbidities.
ANSWER: Causality is not mechanistic. Estrogen metabolites were not mentioned. Partners and co-regulators of ERs always percept the danger of genome and exert efforts for the upregulation of ER activation. Aging, adiposity, or metabolic disorders all are consequences of weakening estrogen signaling instead of confounding variables.
- Figures (if any) are schematic and overly simplistic. They fail to represent molecular details such as receptor dimerization, transcriptional co-activation, or pathway cross-talk. A robust mechanistic review would require well-annotated figures linking estrogen signaling to DNA damage, inflammation, and tumor microenvironment remodeling.
ANSWER: Fig. 1 was concealed as it was simplistic and superfluous indeed, as the details of liganded and unliganded activation of ERs maybe found in the cited works. By contrast, Fig. 2 is very important, as it helps to understand the mechanism of contraction activated glucose uptake in skeletal muscles. This mechanism was not revealed until now, and Fig. 2 represents a novelty in its simple form. A new Fig. 1. was included into chapter 8. This figure shows the molecular changes of deepening insulin resistance and the compensatory efforts for the improvement of estrogen signaling. The defect of estrogen signaling endangers DNA stability, while hyperinsulinemia, androgen and IGF-1 excess, as well as pro-inflammatory cytokine activation all are efforts for increasing aromatase expression and ER activation, rather than for cancer initiation. My article (Ref. 8) describes the double advantage of estrogen signaling, causing apoptotic death in tumors, while an upregulation of immune defense against tumors. Ref. 8. gained a Breakthrough Award from Scifax Company (India) in 2025, following an invited application.
- The tone of writing is unscientific and occasionally polemical. Phrases such as “all cancers arise from estrogen damage” and “modern lifestyle breaks hormonal harmony” read more like advocacy statements than scientific reasoning. The manuscript would benefit from restraint, precision, and evidence-based argumentation.
ANSWER: “estrogen signaling damage” instead of estrogen damage! When insulin resistance is associated with near all kind of tumors according to the present knowledge, the origin of insulin resistance in the background (defect of estrogen signaling) is a prerequisite of cancer initiation. The whole work shows the close correlations between insulin resistance and deficient estrogen signaling, and it is supported by numerous references. It is true that the vast majority of literary data support the estrogen induced breast cancer, and the origin of insulin resistance is unknown. Nevertheless, the theory of “estrogen induced cancer of female organs” is not extended to all cancers, this is a misunderstanding. Conversely, in Chapter 4, the conspicuously high incidence of breast and female cancers among patients with defective estrogen signaling, is explained by their extremely high demand for estrogen driven genomic regulation. I could not find “modern lifestyle breaks hormonal harmony” in the manuscript, but it would be corrected to hormonal equilibrium.
- The discussion ignores well-established counterexamples and contradictions. For instance, estrogen exerts protective effects in certain tissues (e.g., colon, heart, brain). The authors’ one-directional framing of estrogen as uniformly oncogenic is inaccurate and fails to reflect the bidirectional nature of its biological roles.
ANSWER: Uniform carcinogenic activity of estrogens in several organs was not mentioned, it is a misunderstanding. Only breast, endometrium and ovarian tumors are regarded as estrogen induced ones according to the traditional literary data. According to the current belief, ER negative breast cancers are not hormone dependent. However, ER negative breast cancers are not quite different tumors, but they are unable to fight against deregulation attributed to a serious ER resistance. Estrogen may not be bidirectional in its biological roles. Data in schoolbooks comprise that sky-high estrogen level is necessary for ovulation and conception. High estrogen levels orchestrate embryonal development and differentiation of all organs. How can be estrogen carcinogenic for female organs and protective for others? Until now, nobody could answer this question.
- References are poorly curated and imbalanced. Key studies in estrogen receptor structure–function, endocrine disruption mechanisms, and hormone-related carcinogenesis are omitted. Many citations are outdated, secondary sources, or lack DOI verification.
ANSWER: Key studies on estrogen receptor structure and function were referenced, mainly those, which do not force the breast cancer induction. Endocrine disruption is never associated with ovarian hormones, only with synthetic estrogens and progestins. Altered chemical structure of synthetic hormones causes complications in medical practice, typically insulin resistance and increased breast cancer risk. These problems were not included into this manuscript, but maybe found in refs. 140 and 141. Outdated citations comprise important first establishments. Lack of DOI verification is my mistake, I have completed these references.
- The manuscript has no coherent conclusion or future perspective. A credible review should identify gaps in knowledge and propose research directions—e.g., how estrogen signaling interacts with metabolic or immune pathways in specific cancer types—rather than making sweeping universal claims.
ANSWER: Discussion has been completed according to the suggestions of Reviewer 2 in red.
- The English is understandable but stylistically inconsistent, with multiple grammatical errors, imprecise phrasing, and poor paragraph transitions. Substantial language and logic editing would be necessary even after conceptual revision.
ANSWER: The English language of manuscript was edited by MDPI for the sake of Reviewer 2. According to Reviewer 1 and Reviewer 3 “The English is fine and does not require any improvement.” The imprecise phrasing, and poor paragraph transitions were mentioned in general, without concrete examples I can hardly revise.
Thanks for Reviewer 2 the objections and suggested completions.
Prof. Dr. Zsuzsanna Suba
Reviewer 3 Report
Comments and Suggestions for Authors
Paper by Saba is interesting. Lifestyle factors upregulating estrogen signaling, decrease, while downregulating estrogen signaling increases the risk for cancer according to the paper. Reproduction failure is connected to health disturbances. Author concluded that human body maintains its integrity and functional activity via appropriate estrogen signaling. Advantageous lifestyle factors drive genomic machinery via the upregulation of estrogen signaling ensuring the maintenance of human health. It is very important take-home message for generations
Author Response
RESPONSE TO Reviewer 3.
Open Review
Comments and Suggestions for Authors
Paper by Suba is interesting. Lifestyle factors upregulating estrogen signaling, decrease, while downregulating estrogen signaling increases the risk for cancer according to the paper. Reproduction failure is connected to health disturbances. Author concluded that human body maintains its integrity and functional activity via appropriate estrogen signaling. Advantageous lifestyle factors drive genomic machinery via the upregulation of estrogen signaling ensuring the maintenance of human health. It is very important take-home message for generations.
ANSWER: Thanks for understanding, appreciation and hard work for reviewing my manuscript.
Best regards,
Prof. Dr. Zsuzsanna Suba
Round 2
Reviewer 1 Report
Comments and Suggestions for Authors
The authors took all the provided instructions into account.
Reviewer 2 Report
Comments and Suggestions for Authors
The authors have provided detailed and well-structured responses to the issues I raised. I believe the manuscript is now suitable for publication.